# Complete Blood Count Peculiarities in Pregnant SARS-CoV-2-Infected Patients at Term: A Cohort Study

**DOI:** 10.3390/diagnostics12010080

**Published:** 2021-12-30

**Authors:** Roxana Covali, Demetra Socolov, Razvan Socolov, Ioana Pavaleanu, Alexandru Carauleanu, Mona Akad, Vasile Lucian Boiculese, Ana Maria Adam

**Affiliations:** 1Department of Radiology, Biomedical Engineering Faculty, Grigore T. Popa University of Medicine and Pharmacy Iasi, Elena Doamna Obstetrics and Gynecology University Hospital, 700115 Iasi, Romania; 2Department of Obstetrics and Gynecology, Faculty of Medicine, Grigore T. Popa University of Medicine and Pharmacy Iasi, Cuza Voda Obstetrics and Gynecology University Hospital, 700115 Iasi, Romania; socolov@hotmail.com (D.S.); acarauleanu@yahoo.com (A.C.); 3Department of Obstetrics and Gynecology, Faculty of Medicine, Grigore T. Popa University of Medicine and Pharmacy Iasi, Elena Doamna Obstetrics and Gynecology University Hospital, 700115 Iasi, Romania; socolovr@yahoo.com (R.S.); ioana-m-pavaleanu@umfiasi.ro (I.P.); 4Department of Obstetrics and Gynecology, Grigore T. Popa University of Medicine and Pharmacy, 700115 Iasi, Romania; akad.mona@yahoo.com (M.A.); adam.anamaria89@gmail.com (A.M.A.); 5Department of Statistics, Grigore T. Popa University of Medicine and Pharmacy, 700115 Iasi, Romania; lboiculese@gmail.com

**Keywords:** SARS-CoV-2, pregnant patients, complete blood count, infection risk, fetal outcomes

## Abstract

Background: During viral outbreaks, pregnancy poses an increased risk of infection for women. Methods: In a prospective study, all patients admitted for delivery at term to Elena Doamna Obstetrics and Gynecology University Hospital in Iasi, Romania, between 1 April 2020 and 31 December 2020 were included. There were 457 patients, divided into two groups: group 1, SARS-CoV-2-positive patients (*n* = 46) and group 2, SARS-CoV-2-negative patients (*n* = 411). Among other tests, complete blood count was determined upon admittance, and the following values were studied: white blood cell count, lymphocytes, neutrophils, red blood cell count, hemoglobin, mean corpuscular hemoglobin concentration, mean corpuscular hemoglobin, mean corpuscular volume, red blood cell distribution width, hematocrit, platelet count, mean platelet volume, platelet distribution width, plateletcrit, and platelet large cell ratio. Results: in pregnant SARS-CoV-2-infected patients at term, there was a significant decrease in white blood cell, neutrophil, and lymphocyte count, and an increase in mean corpuscular hemoglobin concentration, compared to healthy pregnant women at term, although all still within normal limits. None of the other components of the complete blood count or fetal outcomes studied was significantly influenced by SARS-CoV-2 infection in pregnant patients at term.

## 1. Introduction

During viral outbreaks, pregnancy poses an increased risk for women due to changes to immune function together with adaptive physiological alterations, such as increased oxygen consumption and edema of the respiratory tract [1,2]. Pregnancy, by virtue of its inherent physiological adaptations, would be expected to increase the risk of morbidity associated with COVID-19, particularly owing to a relatively immunocompromised state secondary to alterations within the body’s cell-mediated immune response and inflammatory mechanisms [3,4]. Therefore, it is imperative to closely monitor laboratory parameters, including the white blood cell count, lymphocyte count, and C-reactive protein, along with other imaging features in computed tomography chest scans, to promptly prevent, diagnose, and treat SARS-CoV-2 infection during pregnancy [5,6].

The white blood cell count, neutrophil count, and neutrophil-to-lymphocyte ratio were higher in those with severe or critical illness than in those with mild or moderate illness [7,8,9], but there was still no significant difference in white blood cell count, neutrophil count, neutrophil-to-lymphocyte ratio, immunologic markers, or coagulation and fibrinolysis markers between pregnant women with coronavirus and those without [10].

Yang [11] also found no difference between pregnant SARS-Cov-2-positive and negative patients during the prenatal and postpartum period, as regards the count of white blood cells, neutrophils, and lymphocytes, the ratio of neutrophils to lymphocytes, and the level of C-reactive protein between the confirmed COVID-19 group and the control group, during the prenatal and postpartum period. 

However, according to Wu [5], white blood cell counts were normal or slightly higher than normal, while the lymphocyte counts were normal or slightly lower than normal, before delivery in patients with SARS-Cov-2 infection, while, according to Sun [12], pregnant COVID-19 patients showed significantly lower numbers of blood lymphocytes and higher numbers of neutrophils compared to healthy pregnant patients. However, Vakili [13] reported leukocytosis and an elevated neutrophil ratio in pregnant COVID-19-positive women. On the other hand, according to Areia [14], pregnant women with COVID-19 only differ from other pregnant women by their lower WBC count.

On the other hand, few studies have focused on the red blood cells’ role in SARS-CoV-2 infection. Bergamaschi [15] reported anemia in 61% of patients, mostly mild and due to inflammation, with lower hemoglobin in female patients. The hemoglobin concentration was influenced by red blood cell distribution width, age, lactate dehydrogenase, and the ratio of arterial partial oxygen pressure to inspired oxygen fraction. However, there have been no studies regarding red blood cells in SARS-CoV-2-positive pregnant patients at term so far.

Since the symptoms may be reminiscent of HELLP (hemolysis, elevated liver enzymes, low platelet count) syndrome, the evaluation of platelet count in all pregnant patients presenting with COVID-19, as per International Society on Thrombosis and Hemostasis (ISTH) guidance, is paramount for appropriate diagnosis and management [1].

Due to the lack of consensus between different studies and the relatively low number of studies regarding pregnant patients at term, the aim of this study was to determine any correlation between the presence of SARS-CoV-2 infection in pregnant patients at term and their complete blood count.

## 2. Materials and Methods

In a prospective study, all patients admitted for delivery at term to Elena Doamna Obstetrics and Gynecology University Hospital in Iasi, Romania, between 1 April 2020 (when we were designated a COVID-19 support hospital) and 31 December 2020 were included. Inclusion criteria: patients who delivered at term in our hospital whose blood analyses before delivery were performed in our hospital. Exclusion criteria: patients who had systemic inflammation (rheumatoid arthritis, sarcoidosis) and patients with blood diseases (leukemia) were excluded from the study. The remaining 457 patients were divided into two groups: group (1) SARS-CoV-2-positive patients (*n* = 46), and group (2) SARS-CoV-2-negative patients (*n* = 411). Except for the patients who came from another hospital or from a quarantine zone, with a SARS-CoV-2-positive RT-PCR (real-time polymerase chain reaction) test within the last 14 days, all the other patients were RT-PCR tested upon arrival (Appendix A), kept or delivered separately in an intermediate zone, and based on the RT-PCR result they were admitted to the specialized SARS-CoV-2-positive patients’ area or to the SARS-CoV-2-negative patients’ area. Among other tests, complete blood count was determined upon admittance with MAN-HEMATO Laboratory Equipment. The complete blood count values studied were: WBC (white blood cell count), LYM (lymphocytes), MID (leukocytes, other than lymphocytes and granulocytes, that are in a specific size range), NEUT (neutrophils), RBC (red blood cell count), HGB (hemoglobin), MCHC (mean corpuscular hemoglobin concentration), MCH (mean corpuscular hemoglobin), MCV (mean corpuscular volume), RDW (red blood cell distribution width, with two types: RDW-CV and RDW-SD), HCT (hematocrit), PLT (platelet count), MPV (mean platelet volume), PDW (platelet distribution width), PCT (plateletcrit), and P-LCR (platelet large cell ratio). COVID-19 cases are reported to be mild (no pneumonia or mild pneumonia), severe (dyspnea, respiratory frequency ≥30 breaths/min, oxygen saturation [SpO2] ≤ 93%, a ratio of arterial partial pressure of oxygen to fraction of inspired oxygen [PaO2/FiO2] < 300 mm Hg, and/or lung infiltrates >50% within 24 to 48 h), and critical (defined as respiratory failure, septic shock, and/or multiorgan dysfunction or failure) [16,17,18]. We only reported one patient with cough and moderate shortness of breath; all the other patients had no symptoms; therefore, we classified them as mild forms. Since all patients had a mild form of COVID-19, no analysis could be performed to evaluate the results associated with the severity of the disease. As stated above, except for the patients who came from another hospital or from a quarantine zone with a SARS-CoV-2-positive RT-PCR (real-time polymerase chain reaction) test within the last 14 days, all the other patients were RT-PCR tested upon arrival. Most patients were detected as positive at the RT-PCR test upon arrival; the mean and median values of days of PCR positive before arrival were 2.61 (±3.62) and 1.00 (0.00, 6.00), respectively.

Patients’ age was 17–38 years old in group 1 (positive) and 15–45 years old in group 2 (negative). Mean age, parity, and gestation number were not significantly different between the two groups (*p* = 0.156, 0.441, and 0.143, respectively) (Table 1).

Written informed consent was obtained from all patients. Research Ethics Committee Approval from the Elena Doamna University Hospital was obtained for this study (Number 4/2 April 2020).

Statistical analysis was performed with SPSS version 18 software (PASW Statistics for Windows, Chicago: SPSS Inc., Chicago, IL, USA). For descriptive measures, we computed the mean, standard deviation, median, and quartiles 1 and 3 (because the variables follow a non-normal distribution). Therefore, to compare the data, a nonparametric Mann–Whitney U test was applied. Standard cutoff significance *p* = 0.05 was used to assess the validity of the hypothesis.

## 3. Results

### 3.1. Complete Blood Count

Median and mean values of the complete blood count in the pregnant SARS-CoV-2-positive and negative patients at term are shown in Table 2.

We analyzed these values in regards to age, gestation, and parity, in order to determine whether there was any correlation between a particular age group or gestation number or parity number and more accentuated alterations of the complete blood count values.

### 3.2. WBC Increased over Normal Limits Regarding Age

The number and percentage of pregnant SARS-CoV-2-positive and negative patients with a WBC over the normal limits (10 × 10^9^/L in Romania) at term in different age groups is shown in Table 3. There was no patient with a WBC under the normal limits (leukopenia).

There were two pregnant patients in the SARS-CoV-2-positive group and three in the SARS-CoV-2-negative patients’ group whose WBC data before delivery were unknown.

### 3.3. WBC Increased over Normal Limits Regarding Gestation Number

Number and percentage of patients with WBC increased over normal limits (10 × 10^9^/L in Romania) in pregnant SARS-CoV-2 positive and negative patients at term regarding the gestation number (Table 4), and parity (Table 5) is shown below:

### 3.4. Lymphopenia and Lymphocytosis

Although reported frequently in SARS-CoV-2 infection, lymphopenia, represented by the number of lymphocytes under 1000/μL, was visible in only one of our patients: a 30-year-old belonging to the SARS-CoV-2-negative group. No patient in the study group had lymphopenia. However, we noticed that many patients had a lower number of lymphocytes, under 2000/μL. Considering this limit, we studied how many patients in both groups had lower than half the normal number of lymphocytes (normal values being considered 1000–4000/μL), but still within the normal ranges, and how many had lymphocytosis (over 4000/μL), in terms of the age group (Table 6), gestation number (Table 7), and parity number (Table 8). The one lymphopenia patient was included in these tables in the “decreased LYM” category of patients.

### 3.5. Neutropenia and Neutrophilia

Sometimes reported in SARS-CoV-2 infection, neutropenia, as evidenced by a number of neutrophils under 2000/μL, was visible in no patient in the study group, and in only two patients of the control group, while neutrophilia, represented by a number of neutrophils over 8000/μL, was detected in five patients of the study group and 63 patients of the control group. We studied how many patients in both groups had neutropenia, how many were in the normal range, and how many had neutrophilia by age group (Table 9), gestation number (Table 10), and parity (Table 11).

### 3.6. MCHC

Rarely reported in SARS-CoV-2 infection, variations of MCHC beyond normal values (320–360 g/L) were analyzed. We studied how many patients in both groups had MCHC values in the normal ranges and how many were below or above by age group (Table 12), gestation number (Table 13), and parity (Table 14).

### 3.7. Neonate Outcomes

There was no significant difference between the outcomes of neonates of the two groups (Table 15).

### 3.8. Correlations

There are no correlations between any of the significantly different values of WBC, LYM, NEUT, MCHC and the fetal outcomes (correlation coefficients > 0.05).

## 4. Discussion

There are a few articles regarding red blood cell values in COVID-19 patients. Lanini [19] reported no evidence of a significant decrease in red blood cell count below normal in survivors, with a mild, non-statistically significant anemia at the end of follow-up in non-survivors, variations of MCV within the normal range, and RDW steady in the normal range in survivors, with a mild, non-statistically significant anisocytosis at the end of follow-up in non-survivors. There were no specific results for pregnant patients. We also found no significant difference in pregnant SARS-CoV-2-positive patients compared to negative ones, as regards RBC, HGB, MCH, MCV, RDW-CV, RDW-SD, and HCT; we found a significant increase in MCHC (*p* = 0.022), although it was still within normal limits.

In healthy lungs, the oxygen molecules cross the very thin air–blood barrier and enter the capillary blood, where they bind to hemoglobin and are distributed in the whole body. The air–blood barrier consists of very thin components: a surfactant layer; type I pneumocytes (a squamous cell, with very thin cytoplasm); a basement membrane of alveoli, fused and common with the basement membrane of endothelial cells of the pulmonary capillary; and endothelial cells (again, a squamous cell, with very thin cytoplasm) [20,21]. In the case of pulmonary infection by viruses or microbes, the inflammatory process will cover the alveoli surface with exudate and fill the interstitial spaces between alveoli and pulmonary capillaries with exudate. These are the stages: the excessive release of toxins [22,23] by invading microorganisms generates the disruption of capillary integrity [24], which, in turn, leads to endothelial hyperpermeability and alveolar flooding. This leads to an accumulation of protein-rich fluid in the alveolar space, impairing gas exchange and precipitating respiratory distress [25]; because this exudate thickens the layers to be crossed by the oxygen molecules, fewer oxygen molecules manage to enter the pulmonary capillaries, and fewer are transported by hemoglobin. Due to the decrease in the amount of oxygen reaching the organs, there is an increase in the number of red blood cells and the amount of hemoglobin in the red blood cells, a situation visible in adaptation polyglobulia in persons living at high altitude or in chronically ill cardiac patients. This polyglobulia occurs when the oxygen supply is strongly diminished. However, the pregnant SARS-CoV-2-positive patients included in this study presented only a mild form of the disease, with no respiratory symptoms, and therefore, the amount of exudate in their alveoli must have been small or absent and the decrease in oxygen supply very small, so the body required only a tiny adaptation, in the form of a light increase in MHCH, without the requirement of an increase in the number of red blood cells or in the amount of hemoglobin.

The immunology of pregnancy is a highly controlled system, allowing the survival of the fetus and simultaneously protecting the pregnant woman [26]. During normal pregnancy, there is a significant increase in the absolute numbers of both populations of phagocytes [27]. In pregnant COVID-19 patients, white blood cell count and neutrophil count and percentage were significantly higher, whereas mean lymphocyte percentage was significantly lower compared with nonpregnant COVID-19 patients [28]. Individuals with reduced levels of phagocytes (especially monocytes and neutrophils) are extremely susceptible to and often struggle to recover from infection [29].

Some studies [30,31,32] showed that lymphopenia was the leading laboratory change in pregnant women confirmed with Sars-CoV-2 infection. This was not true in our study: we only found one pregnant patient at term with lymphopenia, which belonged to the pregnant SARS-CoV-2-negative patients, while no pregnant SARS-CoV-2-positive patient at term in our study group had lymphopenia. However, many pregnant patients had a lymphocyte count lower than 2000/μL, which means less than half the normal amount (normal values being considered 1000–4000/μL) number of lymphocytes, but still within the normal range. A lymphocyte count lower than 2000/μL was found in 34.09% of pregnant Sars-CoV-2-positive patients at term (ranging between 10% in 20–24-year-old patients and 60% in 35–39-year-old patients), but only in 17.64% of pregnant Sars-CoV-2-negative patients at term (ranging from 11.53% of <19-year-old patients to 40.54% of 35–39-year-old patients). This lower lymphocyte count varied from 20% of gesta 1 to 50% of gestas 4 and 5 in pregnant Sars-CoV-2-positive patients at term and from 8.33% in gesta 6 to 100% in gesta 11 in pregnant Sars-CoV-2-negative patients at term. It also varied from 10% of paras 3 and 5 to 21% of para 1 in pregnant Sars-CoV-2-positive patients at term and from 5.88% in para 4 to 40% in para 8 in pregnant Sars-CoV-2-negative patients at term. This is according to Wu [5], who noticed that the lymphocyte counts were normal or slightly lower than the normal lower limit before delivery in pregnant SARS-CoV-2-infected women. In SARS-CoV-2-infected patients with respiratory complaints, Aya [33] reported lymphocytopenia in 45.5% and 32% of pregnant and nonpregnant patients, respectively. We reported only one pregnant lymphocytopenia patient in the SARS-CoV-2-negative group, and none in the pregnant SARS-CoV-2-positive patient group at term, and none of them had respiratory complaints. Lymphopenia may be associated with moderate or severe cases of SARS-CoV-2 infection, while our patients had only a mild form of infection and no respiratory complaints. This would be in accordance with Berry [34], who demonstrated that, in pregnant women who tested positive for SARS-CoV-2, significant laboratory findings associated with increasing disease severity included decreased hemoglobin and white blood cell count, lymphopenia, and increasing levels of inflammatory markers.

Yang [11] found that, during the prenatal and postpartum period, there was no difference in the count of WBC, neutrophils, and lymphocytes, the ratio of neutrophils to lymphocytes, or the level of CRP between the confirmed COVID-19 group and the control group (*p* < 0.05). As we have shown above, we found a significant difference in the median values of WBC, lymphocytes, and neutrophils between pregnant SARS-CoV-2-positive and negative patients at term. This was confirmed by Shi [35], who, in an extensive review, demonstrated that the most frequent abnormalities were elevated D-dimer (82%), elevated neutrophil count (81%), elevated C-reactive protein (69%), and decreased lymphocyte count (59%).

This study has several limitations: as other cohort studies noticed [36,37], most patients had only a mild form of infection, and so may not be representative of COVID-19 infection. Therefore, a multicentric study involving many more patients and countries worldwide would be necessary to confirm these results. Second, serum inflammatory markers values, such as cytokines, were not available for this study; therefore, an interesting comparison/correlation between blood cell counts and serum inflammatory markers was not performed, a correlation that might help better understand the SARS-Cov-2 infection. Third, this study was performed during the initial outbreak of the SARS-CoV-2 pandemic, with the original variant of the virus circulating in 2020. However, two new variants emerged in 2021 in Romania: the alpha variant in the spring of 2021 and the delta variant in the summer and autumn of 2021; a comparison of the complete blood count in pregnant patients at term between all three variants of the virus would be a necessary update. Fourth, these results from Romania, for these variants of the virus, may not apply to other populations, which may be infected with other variants of the SARS-CoV-2 virus.

## 5. Conclusions

This study demonstrated that in pregnant SARS-CoV-2-infected patients at term, there was a significant decrease in WBC, NEUT, and LYM and an increase in MHCH, compared to healthy pregnant women at term, although all values were still within normal limits. None of the other components of the complete blood count or fetal outcomes studied were influenced by SARS-CoV-2 infection in pregnant patients at term.

## Figures and Tables

**Table 1 diagnostics-12-00080-t001:** Patient characteristics: mean values (and standard deviation) on the upper line and median values (quartile 1, quartile 2) on the lower line of each value below.

Pregnant Patients at Term	SARS-CoV-2 Positive	SARS-CoV-2 Negative	Significance, *p*
Age (years)	27.83 (±5.28)	26.76 (±6.27)	0.156
	28 (23, 32)	27 (22, 31)	
Gestation (number)	2.28 (±1.3)	2.56 (±1.86)	0.441
	2 (1, 3)	2 (1, 3)	
Parity (number)	1.87 (±1.14)	2.15 (±1.43)	0.143
	2 (1, 3)	2 (1, 3)	

**Table 2 diagnostics-12-00080-t002:** Patient characteristics: mean values (and standard deviation) on the upper line, and median values (quartile 1, quartile 2) on the lower line of each value below, in pregnant SARS-Cov-2-positive patients at term compared to pregnant SARS-CoV-2-negative patients at term.

Median Blood Values (Mean Values)	Pregnant SARS-CoV-2-Positive Patients at Term	Pregnant SARS-CoV-2-Negative Patients at Term	Significance, *p*
WBC	8.34 (±3.30)	10.77 (±2.60)	0.000
	9.60 (7.11, 10.99)	11.07 (9.27, 12.79)	
LYM	2.06 (±0.56)	2.59 (±0.69)	0.000
	2.16 (1.64, 3.00)	2.64 (1.67, 4.14)	
MID	1.99 (±1.20)	2.03 (±1.12)	0.568
	2.19 (1.34, 2.83)	2.25 (1.48, 2.83)	
NEUT	4.86 (±2.42)	5.96 (±1.88)	0.000
	5.25 (3.67, 6.05)	6.17 (4.87, 7.32)	
RBC	4.17 (±0.41)	4.14 (±0.35)	0.554
	4.1 (3.96, 4.42)	4.14 (3.91, 4.35)	
HGB	11.60 (±1.17)	11.70 (±1.16)	0.436
	11.87 (10.90, 12.85)	11.63 (10.90, 12.40)	
MCHC	355.35 (±14.71)	349.68 (±16.55)	0.022
	351.86 (345.58, 359.06)	346.08 (337.21, 357.96)	
MCH	28.52 (±1.89)	28.31 (±19.10)	0.733
	28.36 (26.89, 29.80)	29.59 (26.54, 29.99)	
MCV	80.44 (±5.13)	81.24 (±5.79)	0.215
	80.45 (77.20, 83.09)	81.22 (77.93, 84.85)	
RDW-CV	14.54 (±1.28)	14.47 (±1.72)	0.869
	14.57 (13.70, 15.43)	14.68 (13.60, 15.47)	
RDW-SD	43.70 (±1.55)	43.57 (±3.11)	0.755
	43.74 (42.68, 44.77)	43.63 (42.53, 44,78)	
HCT	33.76 (±3.07)	33.73 (±2.94)	0.924
	33.75 (31.21, 36.11)	33.61 (31.66, 35.49)	
PLT	265.50 (±65.86)	257 (±59.70)	0.865
	267.39 (219, 295)	264.1 (224, 299)	
MPV	8.05 (±1.15)	8.06 (±1.26)	0.805
	8.32 (7.66, 8.67)	8.35 (7.56, 8.78)	
PDW	16.98 (±2.70)	16.97 (±3.17)	0.804
	16.59 (15.10, 18.41)	16.63 (14.79, 18.77)	
PCT	0.22 (±0.05)	0.21 (±0.05)	0.588
	0.21 (0.19, 0.26)	0.21 (0.18, 0.25)	
P-LCR	22.97 (±7.24)	22.71 (±8.11)	0.840
	24.52 (19.22, 27.22)	24.61 (19.17, 28.76)	

WBC = white blood cell count, LYM = lymphocytes, MID = leukocytes, other than lymphocytes and granulocytes, that are in a specific size range, NEUT = neutrophils, RBC = red blood cell count, HGB = hemoglobin, MCHC = mean corpuscular hemoglobin concentration, MCH = mean corpuscular hemoglobin, MCV = mean corpuscular volume, RDW = red blood cell distribution width, with two types (RDW-CV and RDW-SD), HCT = hematocrit, PLT = platelet count, MPV = mean platelet volume, PDW = platelet distribution width, PCT = plateletcrit, P-LCR = platelet large cell ratio.

**Table 3 diagnostics-12-00080-t003:** Number and percentage of pregnant SARS-CoV-2-positive and negative patients with a WBC over the normal limits at term by age group.

	Pregnant SARS-CoV-2-Positive Patients at Term		Pregnant SARS-CoV-2-Negative Patients at Term	
Age (years)	Increased WBC	Normal WBC	Increased WBC	Normal WBC
≤19	0	2	38	14
	(0%)	(100%)	(73.07%)	(26.92%)
20–24	7	3	69	44
	(70%)	(30%)	(61.06%)	(38.93%)
25–29	6	5	71	36
	(54.54%)	(45.45%)	(66.35%)	(33.64%)
30–34	4	15	63	25
	(21.05%)	(78.94%)	(71.59%)	(28.40%)
35–39	0	2	15	22
	(0%)	(100%)	(40.54%)	(59.45%)
≥40	0	0	5	6
	(0%)	(0%)	(45.45%)	(54.54%)
TOTAL	17	27	261	147
	(38.63%)	(61.36%)	(63.97%)	(36.02%)

**Table 4 diagnostics-12-00080-t004:** Number and percentage of pregnant SARS-CoV-2-positive and negative patients with WBC increased over normal limits at term regarding gestation numbers.

	Pregnant SARS-CoV-2-Positive Patients at Term		Pregnant SARS-CoV-2-Negative Patients at Term	
Gestation	Increased WBC	Normal WBC	Increased WBC	Normal WBC
1	9	6	93	31
	60%	40%	75%	25%
2	5	10	77	60
	33.33%	66.66%	56.20%	43.79%
3	2	3	40	31
	40%	60%	56.33%	43.66%
4	1	3	16	13
	25%	75%	55.17%	44.82%
5	0	4	12	6
	0%	100%	66.66%	33.33%
6	0	1	10	2
	0%	100%	83.33%	16.66%
7	0	0	5	2
	0%	0%	71.42%	28.57%
8	0	0	4	0
	0%	0%	100%	0%
9	0	0	2	0
	0%	0%	100%	0%
11	0	0	0	1
	0%	0%	0%	100%
12	0	0	1	1
	0%	0%	50%	50%
15	0	0	1	0
	0%	0%	100%	0%
TOTAL	17	27	261	147
	38.63%	61.36%	63.97%	36.02%

There were no 10, 13, or 14 gestation patients in the groups we studied. There were two pregnant patients in the SARS-CoV-2-positive patient group and three in the SARS-CoV-2-negative patient group whose WBC data before delivery were unknown.

**Table 5 diagnostics-12-00080-t005:** Number and percentage of pregnant SARS-CoV-2-positive and negative patients with a WBC over normal limits at term regarding parity.

	Pregnant SARS-CoV-2-Positive Patients at Term		Pregnant SARS-CoV-2-Negative Patients at Term	
Parity	Increased WBC	Normal WBC	Increased WBC	Normal WBC
1	11	8	112	38
	57.89%	42.10%	74.66%	25.33%
2	5	13	86	63
	27.77%	72.22%	57.71%	42.28%
3	0	2	32	34
	0%	100%	48.48%	51.51%
4	1	2	12	5
	33.33%	66.66%	70.58%	29.41%
5	0	1	7	3
	0%	100%	70%	30%
6	0	1	5	1
	0%	100%	83.33%	16.66%
7	0	0	3	1
	0%	0%	75%	25%
8	0	0	3	2
	0%	0%	60%	40%
9	0	0	0	0
	0%	0%	100%	0%
11	0	0	0	1
	0%	0%	0%	100%
12	0	0	1	0
	0%	0%	100%	0%
15	0	0	0	0
	0%	0%	100%	0%
TOTAL	17	27	261	147
	38.63%	61.36%	63.97%	36.02%

There were no 10, 13, 14, or 15 parity patients in the groups we studied. There were two pregnant patients in the SARS-CoV-2-positive patient group and three in the SARS-CoV-2-negative patient group whose WBC data before delivery were unknown.

**Table 6 diagnostics-12-00080-t006:** Number and percentage of pregnant patients with a normal, increased, and decreased number of lymphocytes in regards to age.

KERRYPNX	Pregnant SARS-CoV-2-Positive Patients at Term			Pregnant SARS-CoV-2-Negative Patients at Term		
Age (Years)	Decrease LYM	Normal LYM	Increased LYM	Decreased LYM	Normal LYM	Increased LYM
≤19	1	1	0	6	41	5
	50%	50%	0%	11.53%	78.84%	9.61%
20–24	1	9	0	17	92	4
	10%	90%	0%	15.04%	81.41%	3.53%
25–29	4	7	0	15	91	1
	36.36%	63.63%	0%	14.01%	85.04%	0.93%
30–34	6	10	0	15	68	5
	37.5%	62.5%	0%	17.04%	77.27%	5.68%
35–39	3	2	0	15	21	1
	60%	40%	0%	40.54%	56.75%	2.70%
≥40	0	0	0	4	7	0
	0%	0%	0%	36.36%	63.63%	0%
TOTAL	15	29	0	72	320	16
	34.09%	65.90%	0%	17.64%	78.43%	3.92%

Decreased = lower than 2000/μL—that is, in the lower half of normal values, not lymphopenia. The only pregnant lymphopenia patient was 19-years-old, had 790/μL lymphocytes, belonged to the SARS-CoV-2-negative group, and was included with the decreased number of lymphocytes patients. There were two pregnant patients, aged 22 and 25 years old, in the SARS-CoV-2-positive patient group and three, aged 28, 28, and 32 years old, respectively, in the SARS-CoV-2-negative patient group whose lymphocyte data before delivery were unknown.

**Table 7 diagnostics-12-00080-t007:** Number and percent of pregnant patients with a normal, increased, and decreased number of lymphocytes, as regards gestation.

	Pregnant SARS-CoV-2-Positive Patients at Term			Pregnant SARS-CoV-2-Negative Patients at Term		
Gestation	Decreased LYM	Normal LYM	Increased LYM	Decreased LYM	Normal LYM	Increased LYM
1	3	12	0	16	97	8
	20%	80%	0%	13.22%	80.16%	6.61%
2	6	9	0	26	108	4
	40%	60%	0%	18.84%	78.26%	2.89%
3	2	3	0	18	49	3
	40%	60%	0%	25.71%	70%	4.28%
4	2	2	0	5	26	0
	50%	50%	0%	16.12%	83.87%	0%
5	2	2	0	3	16	0
	50%	50%	0%	15.78%	84.21%	0%
6	0	1	0	1	11	0
	0%	100%	0%	8.33%	91.66%	0%
7	0	0	0	1	6	0
	0%	0%	0%	14.28%	85.71%	0%
8	0	0	0	0	3	1
	0%	0%	0%	0%	75%	25%
9	0	0	0	1	1	0
	0%	0%	0%	50%	50%	0%
11	0	0	0	1	0	0
	0%	0%	0%	100%	0%	0%
12	0	0	0	0	2	0
	0%	0%	0%	0%	100%	0%
15	0	0	0	0	1	0
	0%	0%	0%	0%	100%	0%
TOTAL	15	29	0	72	320	16
	34.09%	65.90%	0%	17.64%	78.43%	3.92%

Decreased = lower than 2000/μL—that is, in the lower half of normal values, not lymphopenia. The only pregnant lymphopenia patient, gesta 6, had 790/μL lymphocytes, belonged to th SARS-CoV-2-negative group, and was included in the decreased number of lymphocytes group. There were two pregnant patients, gesta 1 and gesta 2, in the SARS-CoV-2-positive group and three, gesta 1, gesta 1, and gesta 4, in the SARS-CoV-2-negative group whose lymphocyte data before delivery were unknown.

**Table 8 diagnostics-12-00080-t008:** Number and percent of pregnant patients with a normal, increased, and decreased number of lymphocytes, as regards parity.

	Pregnant SARS-CoV-2-Positive Patients at Term			Pregnant SARS-CoV-2-Negative Patients at Term		
Parity	Decreased LYM	Normal LYM	Increased LYM	Decreased LYM	Normal LYM	Increased LYM
1	4	15	0	21	120	9
	21.05%	78.94%	0%	14%	80%	6%
2	7	11	0	25	119	5
	38.88%	61.11%	0%	16.77%	79.86%	3.35%
3	2	0	0	20	45	1
	100%	60%	0%	30.30%	68.18%	1.51%
4	1	2	0	1	16	0
	33.33%	66.66%	0%	5.88%	94.11%	0%
5	1	0	0	1	9	0
	100%	0%	0%	10%	90%	0%
6	0	1	0	1	5	0
	0%	100%	0%	16.66%	83.33%	0%
7	0	0	0	1	2	1
	0%	0%	0%	25%	50%	25%
8	0	0	0	2	3	0
	0%	0%	0%	40%	60%	0%
9	0	0	0	0	0	0
	0%	0%	0%	0%	0%	0%
11	0	0	0	0	0	0
	0%	0%	0%	0%	0%	0%
12	0	0	0	0	1	0
	0%	0%	0%	0%	100%	0%
15	0	0	0	0	0	0
	0%	0%	0%	0%	0%	0%
TOTAL	15	29	0	72	320	16
	34.09%	65.90%	0%	17.64%	78.43%	3.92%

Decreased = lower than 2000/μL—that is, in the lower half of normal values, not lymphopenia. The only pregnant lymphopenia patient, para 6, had 790/μL lymphocytes, belonged to the SARS-CoV-2-negative group, and was included in the decreased number of lymphocytes group. There were two pregnant patients, both para 1, in the SARS-CoV-2-positive group and three, para 1, para 1, and para 4, in the SARS-CoV-2-negative group whose lymphocyte data before delivery were unknown.

**Table 9 diagnostics-12-00080-t009:** Number and percent of pregnant patients with a normal, increased (neutrophilia), and decreased (neutropenia) number of neutrophils (NEUT) by age group.

	Pregnant SARS-CoV-2-Positive Patients at Term			Pregnant SARS-CoV-2-Negative Patients at Term		
Age (years)	Decreased NEUT	Normal NEUT	Increased NEUT	Decreased NEUT	Normal NEUT	Increased NEUT
≤19	0	2	0	0	42	10
	0%	100%	0%	0%	80.76%	19.23%
20–24	0	7	3	0	95	18
	0%	70%	30%	0%	84.07%	15.92%
25–29	0	9	2	0	92	15
	0%	81.81%	18.18%	0%	85.98%	14.01%
30–34	0	16	0	2	73	13
	0%	100%	0%	2.27%	82.95%	14.77%
35–39	0	5	0	0	31	6
	0%	100%	0%	0%	83.78%	16.21%
≥40	0	0	0	0	10	1
	0%	0%	0%	0%	90.90%	9.09%
TOTAL	0	39	5	2	343	63
	0%	88.63%	11.36%	0.49%	84.06%	15.44%

Decreased = neutropenia, neutrophils < 2000/μL, increased = neutrophilia, neutrophils > 8000/μL. There were two pregnant patients, aged 22 and 25 years old, in the SARS-CoV-2-positive group and three, aged 28, 28, and 32 years old, respectively, in the SARS-CoV-2-negative group whose neutrophil count before delivery was unknown.

**Table 10 diagnostics-12-00080-t010:** Number and percentage of pregnant patients with a normal, increased, and decreased number of neutrophils, as regards gestation.

	Pregnant SARS-CoV-2-Positive Patients at Term			Pregnant SARS-CoV-2-Negative Patients at Term		
Gestation	Decreased NEUT	Normal NEUT	Increased NEUT	Decreased NEUT	Normal NEUT	Increased NEUT
1	0	12	3	2	98	21
	0%	80%	20%	1.65%	80.99%	17.35%
2	0	14	1	0	117	21
	0%	93.33%	6.66%	0%	84.78%	15.21%
3	0	5	0	0	61	9
	0%	100%	0%	0%	87.14%	12.85%
4	0	3	1	0	25	6
	0%	75%	25%	0%	80.64%	19.35%
5	0	4	0	0	18	1
	0%	100%	0%	0%	94.73%	5.26%
6	0	1	0	0	11	1
	0%	100%	0%	0%	91.66%	8.33%
7	0	0	0	0	6	1
	0%	0%	0%	0%	85.71%	14.28%
8	0	0	0	0	3	1
	0%	0%	0%	0%	75%	25%
9	0	0	0	0	1	1
	0%	0%	0%	0%	50%	50%
11	0	0	0	0	1	0
	0%	0%	0%	0%	100%	0%
12	0	0	0	0	1	1
	0%	0%	0%	0%	50%	50%
15	0	0	0	0	1	0
	0%	0%	0%	0%	100%	0%
TOTAL	0	39	5	2	343	63
	0%	88.63%	11.36%	0.49%	84.06%	15.44%

Decreased = neutropenia, neutrophils < 2000/μL, increased = neutrophilia, neutrophils > 8000/μL. There were two pregnant patients, gesta 1 and gesta 2, in the SARS-CoV-2-positive group and three, gesta 1, gesta 1, and gesta 4, in the SARS-CoV-2-negative group whose neutrophil count before delivery was unknown.

**Table 11 diagnostics-12-00080-t011:** Number and percentage of pregnant patients with a normal, increased, and decreased number of neutrophils (NEUT), as regards parity.

	Pregnant SARS-CoV-2-Positive Patients at Term			Pregnant SARS-CoV-2-Negative Patients at Term		
Parity	Decreased NEUT	Normal NEUT	Increased NEUT	Decreased NEUT	Normal NEUT	Increased NEUT
1	0	15	4	2	122	26
	0%	78.94%	21.05%	1.33%	81.33%	17.33%
2	0	18	0	0	126	23
	0%	100%	0%	0%	84.56%	15.43%
3	0	2	0	0	59	7
	0%	100%	0%	0%	89.39%	10.60%
4	0	2	1	0	15	2
	0%	66.66%	33.33%	0%	88.23%	11.76%
5	0	1	0	0	10	0
	0%	100%	0%	0%	100%	0%
6	0	1	0	0	5	1
	0%	100%	0%	0%	83.33%	16.66%
7	0	0	0	0	2	2
	0%	0%	0%	0%	50%	50%
8	0	0	0	0	4	1
	0%	0%	0%	0%	80%	20%
9	0	0	0	0	0	0
	0%	0%	0%	0%	0%	0%
11	0	0	0	0	0	0
	0%	0%	0%	0%	0%	0%
12	0	0	0	0	0	1
	0%	0%	0%	0%	0%	100%
15	0	0	0	0	0	0
	0%	0%	0%	0%	0%	0%
TOTAL	0	39	5	2	343	63
	0%	88.63%	11.36%	0.49%	84.06%	15.44%

Decreased = neutropenia, neutrophils < 2000/μL, increased = neutrophilia, neutrophils > 8000/μL. There were two pregnant patients, both para 1, in the SARS-CoV-2-positive group and three, para 1, para 1, and para 4, in the SARS-CoV-2-negative group whose neutrophil count before delivery was unknown.

**Table 12 diagnostics-12-00080-t012:** Number and percentage of pregnant patients with normal, increased, and decreased values of MCHC by age group.

	Pregnant SARS-CoV-2-Positive Patients at Term			Pregnant SARS-CoV-2-Negative Patients at Term		
Age (years)	Decreased MCHC	Normal MCHC	Increased MCHC	Decreased MCHC	Normal MCHC	Increased MCHC
≤19	0	1	1	9	38	5
	0%	50%	50%	17.30%	73.07%	9.61%
20–24	1	7	2	9	91	13
	10%	70%	20%	7.96%	80.53%	11.50%
25–29	0	8	3	5	77	25
	0%	72.72%	27.27%	4.67%	71.96%	23.36%
30–34	1	12	3	6	65	17
	0%	80%	20%	6.81%	73.86%	19.31%
35–39	1	3	1	3	22	12
	20%	60%	20%	8.10%	59.45%	32.43%
≥40	0	0	0	2	5	4
	0%	0%	0%	18.18%	45.45%	36.36%
TOTAL	3	31	10	34	298	76
	6.81%	70.45%	22.72%	8.33%	73.03%	18.62%

There were two pregnant patients, aged 22 and 25 years old, in the SARS-CoV-2-positive patient group and three, aged 28, 28, and 32 years old, respectively, in the SARS-CoV-2-negative group whose MCHC values before delivery were unknown.

**Table 13 diagnostics-12-00080-t013:** Number and percentage of pregnant patients with normal, increased, and decreased values of MCHC, as regards gestation.

	Pregnant SARS-CoV-2-Positive Patients at Term			Pregnant SARS-CoV-2-Negative Patients at Term		
Gestation	Decreased MCHC	Normal MCHC	Increased MCHC	Decreased MCHC	Normal MCHC	Increased MCHC
1	0	12	3	10	88	23
	0%	80%	20%	8.26%	72.72%	19.00%
2	1	10	4	14	101	23
	6.25%	66.66%	26.66%	10.14%	73.18%	16.66%
3	1	2	2	3	55	12
	20%	40%	40%	4.28%	78.57%	17.14%
4	1	2	1	0	26	5
	25%	50%	25%	0%	83.87%	16.12%
5	0	4	0	2	13	4
	0%	100%	0%	10.52%	68.42%	21.05%
6	0	1	0	2	5	5
	0%	100%	0%	16.66%	41.66%	41.66%
7	0	0	0	0	4	3
	0%	0%	0%	0%	57.14%	42.85%
8	0	0	0	0	3	1
	0%	0%	0%	0%	75%	25%
9	0	0	0	1	1	0
	0%	0%	0%	50%	50%	0%
11	0	0	0	1	0	0
	0%	0%	0%	100%	0%	0%
12	0	0	0	1	1	0
	0%	0%	0%	50%	50%	0%
15	0	0	0	0	1	0
	0%	0%	0%	0%	100%	0%
TOTAL	3	31	10	34	298	76
	6.81%	70.45%	22.72%	8.33%	73.03%	18.62%

There were two pregnant patients, gesta 1 and gesta 2, in the SARS-CoV-2-positive group and three, gesta 1, gesta 1, and gesta 4, in the SARS-CoV-2-negative group whose neutrophil count before delivery was unknown.

**Table 14 diagnostics-12-00080-t014:** Number and percentage of pregnant patients with normal, increased, and decreased values of MCHC, as regards parity.

	Pregnant SARS-CoV-2-Positive Patients at Term			Pregnant SARS-CoV-2-Negative Patients at Term		
Parity	Decreased MCHC	Normal MCHC	Increased MCHC	Decreased MCHC	Normal MCHC	Increased MCHC
1	0	15	4	13	109	28
	0%	78.94%	21.05%	8.66%	72.66%	18.66%
2	2	11	5	12	110	27
	11.11%	61.11%	27.77%	8.05%	73.82%	18.12%
3	0	1	1	3	55	8
	0%	50%	50%	4.54%	83.33%	12.12%
4	1	2	0	0	13	4
	33.33%	66.66%	0%	0%	76.47%	23.52%
5	0	1	0	4	2	4
	0%	100%	0%	40%	20%	40%
6	0	1	0	0	4	2
	0%	100%	0%	0%	66.66%	33.33%
7	0	0	0	0	2	2
	0%	0%	0%	0%	50%	50%
8	0	0	0	1	3	1
	0%	0%	0%	20%	60%	20%
9	0	0	0	0	0	0
	0%	0%	0%	0%	0%	0%
11	0	0	0	1	0	0
	0%	0%	0%	100%	0%	0%
12	0	0	0	0	0	0
	0%	0%	0%	0%	0%	0%
15	0	0	0	0	0	0
	0%	0%	0%	0%	0%	0%
TOTAL	3	31	10	34	298	76
	0%	88.63%	11.36%	0.49%	84.06%	15.44%

There were two pregnant patients, both para 1, in the SARS-CoV-2-positive group and three, para 1, para 1, and para 4, in the SARS-CoV-2-negative group whose neutrophil count before delivery was unknown.

**Table 15 diagnostics-12-00080-t015:** Neonate characteristics: mean values (and standard deviation) on the upper line, and median values (quartile 1, quartile 2) on the lower line of each value below, in neonates from pregnant SARS-Cov-2-positive patients at term compared to neonates from pregnant SARS-CoV-2-negative patients at term.

Mothers Neonates	SARS-CoV-2-Positive Patients at Term (*n* = 46)	SARS-CoV-2-Negative Patients at Term (*n* = 411)	*p*
Weight	3.37 (±0.61)	3.36 (±0.43)	0.67
	3.42 (2.98, 3.77)	3.34 (3.06, 3.68)	
Apgar score	8.28 (±0.93)	8.49 (±0.64)	0.18
	8.00 (8.00, 9.00)	9.00 (8.00, 9.00)	
Gender: male	25 (54.34%)	218 (53.04%)	0.86

## Data Availability

All the data are available from the corresponding author upon reasonable request.

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
