# Peer review of "Complete Blood Count Peculiarities in Pregnant SARS-CoV-2-Infected Patients at Term: A Cohort Study"

_diagnostics, 2021, doi:10.3390/diagnostics12010080_

Round 1
Reviewer 1 Report
Exclusion criteria:
The aforementioned exclusion criteria are actually inclusion criteria not met, I suggest that the authors treat the exclusion criteria as those confounding variables with which they want to avoid bias. As it is currently postulated, the exclusion criteria are redundant if the inclusion criteria are met, the exclusion criteria must be variables that could be present even when the inclusion criteria were met, for example: patients with evidence of chorioamnioitis (thinking that chorioamnioitis can modify the white cell count and confuse the effect of the SARS-COV-2 infection)
Results:
They are shown both in the text and in Table 2, which is redundant, the authors must choose the alternative to present them.
Justify, what would be the justification for stratifying results by age, parity, and gestation number?
In the clinical context, were there any analyzes that evaluated the results associated with the severity of the disease?
Author Response
Response to Reviewer 1
Open Review
(x) I would not like to sign my review report
( ) I would like to sign my review report
English language and style
( ) Extensive editing of English language and style required
( ) Moderate English changes required
( ) English language and style are fine/minor spell check required
(x) I don't feel qualified to judge about the English language and style
Yes |
Can be improved |
Must be improved |
Not applicable |
|
Does the introduction provide sufficient background and include all relevant references? |
(x) |
( ) |
( ) |
( ) |
Is the research design appropriate? |
(x) |
( ) |
( ) |
( ) |
Are the methods adequately described? |
( ) |
(x) |
( ) |
( ) |
Are the results clearly presented? |
( ) |
(x) |
( ) |
( ) |
Are the conclusions supported by the results? |
(x) |
( ) |
( ) |
( ) |
Comments and Suggestions for Authors
Exclusion criteria:
The aforementioned exclusion criteria are actually inclusion criteria not met, I suggest that the authors treat the exclusion criteria as those confounding variables with which they want to avoid bias. As it is currently postulated, the exclusion criteria are redundant if the inclusion criteria are met, the exclusion criteria must be variables that could be present even when the inclusion criteria were met, for example: patients with evidence of chorioamnioitis (thinking that chorioamnioitis can modify the white cell count and confuse the effect of the SARS-COV-2 infection.
-We DELETED the initial Exclusion criteria, as follows:
patients who delivered elsewhere and were afterwards admitted to our hospital; patients who delivered in our hospital as soon as they arrived, so that blood harvest for analysis before delivery could not be performed; patients who had the blood analysis performed in another hospital and were then rushed to our hospital for delivery (because we are a COVID-19 support hospital, and others are not) were excluded from the study.
And we replaced it with lines 100-102:
Exclusion criteria: patients who had systemic inflammation (rheumatoid arthritis, sarcoidosis), patients with blood diseases (leukemia) were excluded from the study.
Results:
They are shown both in the text and in Table 2, which is redundant, the authors must choose the alternative to present them.
-We DELETED THE TEXT, shown below, and we left there Table 2, which is more visible and easy to follow:
Mann–Whitney test showed that there was no significant difference in the median values of MID, RBC, HGB, MCH, MCV, RDW-CV, RDW-SD, HCT, PLT, MPV, PDW, PCT, and P-LCR between pregnant SARS-CoV-2-positive and -negative patients at term (P = 0.568, 0.554, 0.436, 0.733, 0.215, 0.869, 0.755, 0.924, 0.865, 0.805, 0.804, 588, and 0.840, respectively). Still, there was a significant difference between the median values of WBC, LYM, NEUT, and MCHC between pregnant SARS-CoV-2-positive and -negative patients at term (P = 0.00, 0.00, 0.00, and 0.022, respectively).
Justify, what would be the justification for stratifying results by age, parity, and gestation number?
-We wrote, in lines 167-169:
We analyzed these values as regards age, gestation and parity, in order to determine whether there was any correlation between a particular age group or gestation number or parity number and more accentuated alterations of the complete blood count values.
In the clinical context, were there any analyzes that evaluated the results associated with the severity of the disease?
-We wrote, in lines 121-130:
COVID-19 cases are reported to be mild (no pneumonia or mild pneumonia), severe (dyspnea, respiratory frequency ≥30 breaths/min, oxygen saturation [SpO2] ≤93%, a ratio of arterial partial pressure of oxygen to fraction of inspired oxygen [PaO2/FiO2] <300 mm Hg, and/or lung infiltrates >50% within 24 to 48 hours), and critical (defined as respiratory failure, septic shock, and/or multiorgan dysfunction or failure). [16-18]. We only report one patient with cough and moderate shortness of breath, all the other patients had no symptoms, therefore we classified them as mild forms. Since all patients had a mild form of COVID-19, there could be performed no analysis to evaluate the results associated with the severity of disease.
Submission Date
22 November 2021
Date of this review
26 Nov 2021 17:40:02

Reviewer 2 Report
More detailed clinical features of studied patients should be presented in Materials and methods section (very briefly mild form is mentioned only in Discussion). Especially, how long was PCR(+) before admittance, what were the clinical COVID-19 symptoms. If possible it is also better to compare blood cell counts with some inflammatory serum markers, cytokines, etc. This can give better understanding of SARS-CoV-2 infection. It should be very interesting to compare obtained data with the consequences for mothr and child after delivery.
Author Response
Response to Reviewer 2
Open Review
(x) I would not like to sign my review report
( ) I would like to sign my review report
English language and style
( ) Extensive editing of English language and style required
( ) Moderate English changes required
( ) English language and style are fine/minor spell check required
(x) I don't feel qualified to judge about the English language and style
Yes |
Can be improved |
Must be improved |
Not applicable |
|
Does the introduction provide sufficient background and include all relevant references? |
(x) |
( ) |
( ) |
( ) |
Is the research design appropriate? |
( ) |
(x) |
( ) |
( ) |
Are the methods adequately described? |
(x) |
( ) |
( ) |
( ) |
Are the results clearly presented? |
( ) |
(x) |
( ) |
( ) |
Are the conclusions supported by the results? |
(x) |
( ) |
( ) |
( ) |
Comments and Suggestions for Authors
More detailed clinical features of studied patients should be presented in Materials and methods section (very briefly mild form is mentioned only in Discussion).
-We added, in lines 121-128:
COVID-19 cases are reported to be mild (no pneumonia or mild pneumonia), severe (dyspnea, respiratory frequency ≥30 breaths/min, oxygen saturation [SpO2] ≤93%, a ratio of arterial partial pressure of oxygen to fraction of inspired oxygen [PaO2/FiO2] <300 mm Hg, and/or lung infiltrates >50% within 24 to 48 hours), and critical (defined as respiratory failure, septic shock, and/or multiorgan dysfunction or failure). [ 16-18]. We only report one patient with cough and moderate shortness of breath, all the other patients had no symptoms, therefore we classified them as mild forms.
Especially, how long was PCR(+) before admittance, what were the clinical COVID-19 symptoms.
-We added :
As stated above, except for the patients who came from another hospital or from a quarantine zone, with a SARS-CoV-2-positive RT-PCR (real-time polymerase chain reaction) test within the last 14 days, all the other patients were RT-PCR tested upon arrival. Most patients were detected as positive at the RT-PCR test upon arrival; the mean and median values of days of PCR positive before arrival were 2.61 (± 3.62) respectively 1.00 (0.00, 6.00).
If possible it is also better to compare blood cell counts with some inflammatory serum markers, cytokines, etc. This can give better understanding of SARS-CoV-2 infection. It would be very interesting to determine whether there is any correlation betwen the inflammatory markers and
-Actually, our laboratory cannot determine cytokines. C-reactive protein was determined in only 3 cases, and two of them were slightly over limits, while erythrocyte sedimentation rate was determined in another one case, so that we cannot correlate these inflammatory markers. Therefore we wrote, in Discussion, lines 377-380:
„Second, serum inflammatory markers values, like cytokines, were not avalable for this study, therefore an interesting comparison /correlation between blood cell counts and serum inflammatory markers was not performed, correlation that might help better understanding the SARS-Cov-2 infection.”
It should be very interesting to compare obtained data with the consequences for mother and child after delivery.
-We added, in lines 285-294:
3.7. Neonate outcomes
There was no significant difference between the outcomes of neonates of the two groups (Table 15).
Table 15. Neonate characteristics: mean values (and standard deviation) on the upper line, and median values (quartile 1, quartile 2) on the lower line of each value below, in neonates from pregnant SARS-Cov-2-positive patients at term compared to neonates from pregnant SARS-CoV-2-negative patients at term.
Mothers Neonates |
SARS-CoV-2-positive patients at term (n=46) |
SARS-CoV-2-negative patients at term (n=411) |
P |
Weight |
3.37 (± 0.61) 3.42 (2.98, 3.77) |
3.36 (± 0.43) 3.34 (3.06, 3.68) |
0.67 |
Apgar score |
8.28 (± 0.93) 8.00 (8.00, 9.00) |
8.49 (± 0.64) 9.00 (8.00, 9.00) |
0.18 |
Gender: male |
25 (54.34%) |
218 (53.04%) |
0.86 |
3.8. Correlations
There are no correlations between any of the significantly different values of WBC, LYM, NEUT, MCHC and the fetal outcomes (correlation coefficients >0.05).
We also added, in Conclusions, (and in Conclusions of the Abstract), as highlighted in yellow:
„None of the other components of the complete blood count or fetal outcomes studied were influenced by SARS-CoV-2 infection in pregnant patients at term.”
Submission Date
22 November 2021
Date of this review
15 Dec 2021 09:51:37
